# Effect of the Thermal Insulation Cover Curing on Temperature Rise and Early-Age Strength of Concrete

**DOI:** 10.3390/ma15082781

**Published:** 2022-04-10

**Authors:** Yuzhang Liu, Jun Zhang, Jiang Chang, Shixiang Xie, Yongsheng Zhao

**Affiliations:** 1Department of Civil Engineering, Tsinghua University, Beijing 100084, China; liuyuzha18@mails.tsinghua.edu.cn (Y.L.); changj20@mails.tsinghua.edu.cn (J.C.); jsx19@mails.tsinghua.edu.cn (S.X.); 2Beijing Uni-Construction Group Company, Ltd., Beijing 100101, China; zhaoys@163.com

**Keywords:** precast concrete, thermal-insulation-cover curing, temperature rise, strength, pore structure

## Abstract

In the present paper, the effects of thermal insulation cover curing on the rise in temperature of concrete, and further on the mechanical properties of the concrete are measured experimentally. In the experiments, polyurethane foam boards of 100 mm thickness were used for thermal insulation, and three concrete strength grades of C30, C50 and C80, were used as concrete samples. Conventional standard curing, and two heat curing methods with a constant temperature of 40 °C and 60 °C were used for comparison. The pore structure of the cement paste subject to the different curing procedures, including insulation cover curing, heat curing and standard curing, was experimentally measured using mercury intrusion porosimetry. The test results show that rate of increase in compressive strength compared with standard curing is 22–34%, 16–26%, and 23–67%, respectively, for C30, C50, C80 concrete after being subject to 1 to 3 days thermal insulation cover curing in the initial period after concrete casting. As expected, initial heat curing of concrete will result in a reduction in the long-term strength of the concrete. At 28 days, the strength reduction rate compared with standard curing due to the insulation cover curing is 3.1–5.9%, 0.6–3.0%, 0–3.2%, respectively, for C30, C50, C80 concrete. By contrast, the compressive strength reduction compared with standard curing at 28 days is 8.6–10.5%, 8.6–9.1%, 4.7–5.6%, respectively, for C30, C50, C80 concrete after being subject to a constant heat curing of 40 °C and 60 °C in the initial period after concrete casting. Measurement of the pore structure of the paste subject to different curing procedures initially after casting shows that rising curing temperature leads to coarser pores, especially an increase in capillary pore fraction. Among the four curing methods used in the present study, the effect of insulation cover curing and low temperature (40 °C) heat curing on the capillary pore content is small, while the effect of high temperature (60 °C) heat curing is significant.

## 1. Introduction

As a widely used construction material, the production of concrete members or structures is normally divided into cast-in-place and cast-in-plant. In concrete production, curing of concrete is normally necessary to attain the required performance of the concrete, whether the concrete is produced on site or in the plant. In particular, for concrete member cast-in-plant, the purpose of curing of concrete includes speeding up the stripping of forms, increasing strength of the elements and shortening the stay-age in plant [1]. To reach the above aims in pre-cast concrete plants, the conventional method is accelerated curing by means of increased temperature and humidity according to economy, availability and long-term performance of materials [2]. Various methods of fast curing have been developed including steam curing at atmospheric pressure, steam curing at elevated pressure, electrical heating and microwave heating [2]. Among these, steam curing under lower water pressure is most generally utilized in pre-cast concrete production. A typical steam curing regime used in a plant consists of 2–4 h precuring after surface finishing, a temperature rising period from environment to 60 °C at the rate of 20 °C per hour, a period with constant temperature for 4–6 h according to environmental temperature, and period of decreasing temperature to environmental temperature at the rate of 20 °C per hour. Existing results showed that an elevated curing temperature results in a coarser, more continuous pore structure for hydrated cement paste [3,4,5,6]. In addition, fast-heating externally should result in inhomogeneous thermal deformation inside the concrete elements that results in thermal stress as well [7,8,9,10,11]. Thermal stress is another source leading to cracking in early age concrete members [12,13]. On the other hand, steam curing used in pre-cast concrete plants obviously increases the cost of the concrete production and the environmental load due to the generation of water steam using coal. In view of sustainability and greenability of concrete production, improvements in curing methods of pre-cast concrete production are still desired.

In concrete, the reaction between Portland cement and water is an exothermic process. If adiabatic conditions between the test sample and environment are satisfied, the temperature inside the concrete sample should gradually increase with hydration age, and finally go into a steady state with a nearly constant temperature. In all other cases, the hydration heat will be lost gradually and the temperature inside the concrete should tend to the temperature of the environment. The amount of cement hydration heat generated in the concrete element depends on the cement amount and/or the concrete volume, water to cement ratio, cement hydration degree and its development with age, and addition of cementitious materials used in the concrete. The temperature rise inside the concrete is dependent on the heat generation and insulation condition of the boundary between concrete and the environment [11]. For example, the temperature rise of the concrete may reach 30 °C, 45 °C and 50 °C, respectively, for concrete with water to binder ratio of 0.3, 0.4 and 0.6 under adiabatic conditions [14,15]. In concrete production, cement hydration heat is generally lost to the surrounding environment without being used. Curing concrete using cement hydration heat may be of great interest either for cast-in-site concrete or for pre-cast concrete production.

In the present paper, cement hydration heat is used as a curing heat resource in concrete production by putting a heat insulation cover on the surface of concrete specimens. The effect of the insulation cover curing initially after concrete casting on the temperature rise of the concrete, and further on compressive strength of the concrete were experimentally studied. Comparative curing methods, including standard curing according to the Chinese standard GB/T 50081-2019 [16] and traditional heat curing with a constant temperature were conducted as well on the same concrete in order to compare with the insulation cover curing. Pore tests on cement paste were used to evaluate the effect of temperature rise on pore structures of the cement paste. To study the effect of concrete strength grade or water to binder ratio, three types of concrete, C30, C50 and C80 concrete were used in the test programs.

## 2. Experimental Programs

### 2.1. Materials and Mix Proportions

To investigate the effect of concrete strength or water to binder ratio on temperature rise under insulation conditions, three concrete mixtures with water to binder ratio (W/B) of 0.62, 0.43 and 0.30 to form three kinds of concretes with compressive strength at 28 days around 30 MPa, 50 MPa and 80 MPa, which are labeled as C30, C50 and C80, respectively, were used in the experiments. All mixtures were made with the same Portland cement. For C30 and C50 concrete, 20% fly ash of the total binder weight was used. For C80 concrete, 10% silica fume of the total binder weight was used. Oxide compositions of the cement, fly ash and silica fume, as well as the mineral compounds of the cement, are listed in Table 1.

Natural sand and crushed limestone with a maximum particle size of 5 mm and 20 mm, respectively, were used as normal fine and coarse aggregates. The concrete mixture proportions used in the present work are listed in Table 2. A polycarboxylate based superplasticizer was used in these mixtures to guarantee that the fresh concrete has a similar slump in 90–120 mm. 

### 2.2. Specimens, Curing and Test Measurements

In the tests, the mold used to store concrete specimens was made of polyurethane board with thickness of 100 mm. Thermal conductivity of the polyurethane board is 0.030175 W/(m K). A plastic mold that can cast three 100 mm^3^ cube specimens was used to cast concrete. The processes of the present heat insulation curing are presented in Figure 1. First, rectangular polyurethane board boxes with side thickness of 100 mm were made. Polyurethane adhesive was used to glue the board connections. In each box, the plastic mold with three cube concrete specimens can be stored. After concrete casting, vibrating and finishing of the casting surface, the steel model was immediately moved into the boxes and the top board was put on and glued to the side boards. The insulation curing was started. To monitor temperature and relative humidity inside the box, a temperature and humidity combined wireless sensor was put into the box before it was closed. According to the time of concrete casting, the concrete was cured inside the box until the age of compressive strength testing. In the present work, the strength was measured after insulation curing of 1, 2 and 3 days. Meanwhile, some of the specimens after insulation curing at the specific age were moved to a standard curing room (temperature of 20 °C; relative humidity greater than 96%) until 28 days after casting. Specimens subject to insulation cover curing were labeled as C.

To make a comparative study with conventional heat curing of concrete, some concrete specimens were made and cured according to the following schedule. After casting, vibrating and finishing, the mold was stored in a room for 1 day. Then the specimen was moved into a heater to raise the temperature to 40 °C and 60 °C, respectively, for heat curing of 1, 2, 3 and 6 days. After that, specimens were demolded and moved into the standard curing room for another certain number of days until the total age is 28 days before compressive strength testing. In the present study, specimens subject to 40 °C and 60 °C heat curing are labeled as LH and HH, respectively, for convenience of analyzing. In addition, specimens under standard curing (temperature of 20 ± 2 °C; relative humidity greater than 96%) [16] were also prepared as a reference. The specimens subject to standard curing were labeled as S. In summary, four curing procedures were used in the test program; they are S-standard curing; C-insulation cover curing; LH-heat curing with constant temperature of 40 °C; HH-heat curing with constant temperature of 60 °C. Here the number age subject to heat curing is the actual age of concrete that includes 1 day staying in the casting room before heating.

To investigate the effect of curing temperature in the initial period after concrete casting on pore structure of the cement paste, cement paste of each concrete (without sand and stone) was prepared and cured as well as the concrete used. The pore structures of the cement paste were measured by mercury intrusion porosimetry (MIP). The test pressure range was 0.0036–413.370 MPa, and the minimum pore diameter was 3.2 nm. At specific age of the curing, some particles with a diameter of 10–15 mm were obtained from the paste sample of the concrete. The samples were placed in absolute ethanol for 24 h and then in an oven at 40 °C to a constant weight to stop hydration.

## 3. Experimental Results and Discussion

### 3.1. Temperature Rise of Concrete under Insulation Curing

Figure 2 shows the experimental results of variation of temperature and relative humidity in the curing box with age of C30, C50 and C80 concrete, respectively. First, it can be observed that the temperature in the box first increases gradually with age starting at the initial environmental temperature of 25.5 °C, 25.8 °C and 26.6 °C, respectively, for C30, C50 and C80 concrete. At age of 25 to 27 h since concrete casting, a peak temperature of 34.2 °C, 38.9 °C and 45.5 °C for C30, C50 and C80 concrete is reached, respectively. Under the designed thickness of the insulation layer and environmental condition, the temperature rise is 10.5 °C, 15.2 °C and 20.3 °C, respectively, for C30, C50 and C80 concrete. After that, the temperature starts to decrease gradually with age and finally returns close or equal to the temperature of the environment. It is noted that the temperature decreasing speed is slower than that of the temperature rise. Under the present simulation conditions, the time of temperature closing to the environmental temperature is 136.6 h, 136.8 h and 93.3 h, respectively for C30, C50 and C80 concrete. Clearly, the temperature inside of the curing box is obviously raised compared to that of the environment due to the insulation action on cement hydration heat. The cement hydration heat is effectively preserved and is used to heat the concrete. Meanwhile, the hydration heat is used to increase cement hydration and concrete strength that is favorable as well for concrete production in pre-cast concrete plants. Second, the rising rate of temperature in the box is less than 1 °C/h, which is much slower than that (20 °C/h) used in steam curing in pre-cast concrete plants. Therefore, possible damage to cement paste due to fast temperature rising should be reduced for the present insulation curing compared to that of traditional steam curing of concrete. This issue should be discussed in the subsequent section based on mercury intrusion measurement of cement paste under different curing procedures. Third, internal relative humidity inside of box can be retained at a relatively high level, especially for C30 and C50 concrete (above or close to 90% within 7 days). For C80 concrete, as expected, relative humidity inside the box starts to drop at about 30 h from initial 85% to 60% at 7 days of the end of the test. The decrease in internal relative humidity inside the box of C80 concrete is due to the higher self-desiccation of high strength concrete or low water to binder ratio of the cement paste. This effect of internal relative humidity in the curing box of high strength concrete on mechanical properties of the concrete should be an interesting query if a long curing period is used. 

### 3.2. Concrete Strength of Concrete under Insulation Curing

Now, it should be interesting to look at the effect of the temperature rise of concrete under insulation condition on the compressive strength of the concrete. Figure 3 displays the compressive strength of concrete at different ages under the designed curing procedures of the present study, including standard curing (S), insulation cover curing (C) and heat curing (low temperature of 40 °C, LH; high temperature of 60 °C, HH). The number after the curing method in the legend of the figure is the corresponding age. It should be noted that only the strength at 28 days is presented for the concrete with constant temperature curing at its early age to study the possible damage of initial heating on concrete. In addition, the purpose of the increased number of testing ages as selected within the initial few days is to explore the possibility of insulation cover curing to replace traditional steam curing for fast remolding in pre-cast concrete pant. 

From Figure 3, it can be seen that after underdoing insulation curing, the compressive strength of concrete at an early age is obviously increased. The benefit of higher early age strength is obtained by simply putting an insulation caver on concrete specimens. For C30 concrete, compared with standard curing (S), the compressive strength is increased from 5.9 to 7.9 MPa (34%), 10.5 to 12.8 MPa (22%), 13.1 to 16.3 MPa (24%), respectively after insulation curing of 1, 2, 3 days. The data in brackets are the rates of increase based on standard curing. At 7 days, the strength of concrete subject to insulation curing is still higher than that of the specimens under standard curing. At 28 days, the compressive strength of concrete subject to insulation curing of 1, 2 and 3 days starting from concrete cast is 34.0 MPa, 33.4 MPa and 34.4 MPa, while the strength of concrete under standard curing is 35.5 MPa. The strength of the former is a bit lower than the latter. The reduction ratio is 4.2%, 5.9% and 3.1%, respectively, for insulation curing of 1, 2 and 3 days at the beginning. By contrast, compressive strength at 28 days of the specimens subject to 40 °C (2 days) and 60 °C (2 days) heat curing becomes 32.4 MPa and 31.8 MPa, respectively. The reduction ratio is 8.9% and 10.5%, which is obviously higher than that of the insulation curing. This means the present insulation curing can increase the early age strength, while the reduction in long term strength due to the initial short term heat curing can be decreased. This positive effect compared with traditional heat curing on concrete is clearly shown in Figure 3a.

For C50 concrete (Figure 3b), a similar trend in the compressive strength of concrete under different curing methods can be observed. Compared with standard curing (S), the compressive strength is increased from 11.7 MPa to 14.5 MPa (24%), 20.5 MPa to 25.8 MPa (26%), 26.1 MPa to 30.3 MPa (16%), respectively after insulation curing (C) of 1, 2, 3 days. As with C30 concrete, at 7 days, the strength of concrete subject to insulation curing is still higher than that of the specimens under standard curing. At 28 days, the compressive strength of concrete subject to insulation curing of 1, 2 and 3 days starting from concrete cast is 57.8 MPa, 56.4 MPa and 56.8 MPa, while the strength of the concrete under standard curing is 58.1 MPa. The strength of the former is a bit lower than that of the latter. The reduction ratio is 0.6%, 3.0% and 2.2%, respectively for insulation curing of 1, 2 and 3 days at the beginning. By contrast, the compressive strength at 28 days of the specimens subject to 40 °C (2 days) and 60 °C (2 days) heat curing becomes 53.1 MPa and 52.8 MPa, respectively. The reduction ratio is 8.6% and 9.1%, which are obviously higher than that of the insulation curing.

For C80 concrete (Figure 3b), compared with standard curing (S), the compressive strength is increased from 24.3 to 40.5 MPa (67%), 42.2 to 56.1 MPa (33%), 47.5 to 58.2 MPa (23%), respectively after insulation curing (C) of 1, 2, 3 days. Similar to C30 and C50, at 7 days, the strength of the concrete subject to insulation curing is still higher than that of the specimens under standard curing. At 28 days, the compressive strength of concrete subject to insulation curing of 1, 2 and 3 days starting from concrete cast is 86.2 MPa, 83.4 MPa and 81.4 MPa, while the strength of the concrete under standard curing is 85.5 MPa. Clearly, the concrete subject to 1-day insulation curing has slightly higher compressive strength than the concrete under standard curing, although the rate of increase is only 0.8%. The other two curing procedures, 2 and 3-day insulation curing, display decreasing compressive strength compared with that of standard curing. The rate of decrease is 2.5% and 3.2%, respectively for 2 and 3-day insulation curing. By contrast, the compressive strength at 28 days of the specimens subject to 40 °C (2 days) and 60 °C (2 days) heat curing becomes 81.5 MPa and 80.7 MPa, respectively. The reduction ratio is 4.7% and 5.6%, respectively for the 40 °C and 60 °C 2-day initial curing.

From the test results showed in Figure 3, it can be concluded that very early-age strength (within the initial 3 days after casting) of concrete can be increased by simply putting on an insulation cover of polyurethane boards. Cement hydration heat can be used for curing concrete, and it should be an increasing choice for reducing or replacing steam curing in pre-cast concrete plants. The rate of increase in early-age strength of insulation curing is influenced by the concrete strength grade or water to binder ratio of the concrete. On the other hand, a negative effect of heat curing on the concrete strength in the long term is observed as well. This phenomenon is observed as well in previous studies [2,5,17,18,19]. The reduction rate is influenced by the listing time of the insulation curing and concrete strength. Based on the present study, the shorter the insulation curing period, the more the reduction on long-term strength is reduced. A rational period of heat curing should be selected based on the requirements of early-age strength and long-term performance. The mechanism of the effect of curing temperature on concrete strength should be discussed in the following section based on pore structure measurement, in turn based on the mercury intrusion method.

### 3.3. Pore Structures of Concrete under Different Curing Procedures

Mercury intrusion porosimetry is a widely used technique to evaluate the pore structures of cementitious materials, including pore size distribution and pore volume fraction. Figure 4 shows a typical pore distribution in terms of cumulative (below) and derivative (above) pore volume versus pore diameter based on MIP of the cement paste of C30 concrete. It should be noted that the pore size here refers to percolation size of the pore corresponding to the intruded pressure of mercury. Although the pores in cement paste may be not ideal round tubes, pore size or diameter measured by MIP can well be used to reflect the equivalent pore size in the paste. From Figure 4, the process of mercury intrusion into cement paste can roughly be divided into three stages according to the rate of the intrusion (dV/dLogD): stage I—initial slow increase stage, relatively large pore diameter variation with small increase in pore volume, the corresponding diameter boundary is (d1, d2) and the pore volume at end of this stage is V1; stage II–rapid increase stage (first peak of dV/dLogD), relatively small pore diameter variation with relatively large increase in pore volume, the corresponding diameter boundary is (d2, d3) and the pore volume at end of this stage is V2; stage III–relatively fast increase stage (second peak of dV/dLogD, normally smaller than the first peak), relatively large pore diameter variation with relatively large increase in pore volume, the corresponding diameter boundary is (d3, d4) and the pore volume at end of this stage is V3. The main contribution of the pores in stage I can be assumed as small amount of gas pores and/or relatively larger capillary pores due to mixing of the fresh paste. For stage II, the contribution of the pores should be the principal amount of the capillary pores, and the ascending and descending of the pore volume rate (dV/dLogD) should be the balance between the number and diameter of the pores. In stage III, the main contribution of the pores should be micropores of C-S-H gel plus a small amount of the capillary pores with narrow size. Therefore, the volume of capillary pore volume can be estimated by V2, and the gel pore volume can be estimated by (V3–V2). A similar classification was used as well by Bahafid et al. [6] for pore characteristic evaluation of well cement under different hydration temperatures. According to the above division method, the effects of different curing procedures on pore structures of the cement paste of C30, C50 and C80 concrete are analyzed below. 

Figure 5, Figure 6 and Figure 7 present pore distribution of the cement pastes subject to different curing procedures designed by the present study in terms of cumulative (below) and derivative (above) pore volume versus pore diameter based on MIP of C30, C50 and C80 concrete, respectively. Again, the number after the curing method shown in the legend is the corresponding age. As an example, pore structure of the cement paste of C30 concrete under different curing procedure is analyzed first in the following.

From Figure 5a, the variation law of pore distribution of the cement paste of C30 concrete under standard curing is clearly shown. A two-peak mode of pore volume rate of increase is observed in the derivative curves. Correspondingly, a two-stage mode of pore volume development is observed as well in the cumulative curves. In stage I, the pore volume (V1) is similar for all pastes cured at different curing age, and d2 is gradually moved forward to the direction of smaller pore size with the increase in curing age. In stage II, the rate peak is gradually moved forward to the smaller pore direction, and the peak value is decreased with the increase in curing age. Correspondingly, the pore volume of V2 and the boundary pore size of diameter d3 are reduced with age due to hydration of the cement. In stage III, corresponding to the second mercury intrusion perk, the pore volume of V3 is significantly decreased with increase in curing age. At 28 days, the total pore volume of the paste of C30 concrete under standard curing is 0.233 mL/g, and the corresponding capillary pore (V2) and gel pore (V3–V2) volumes are 0.125 mL/g (53.6% of the total pore) and 0.108 mL/g (46.4% of the total pore). The pore size of d3 is 13.72 nm which is close to the low limit (about 10 nm) of the general definition of capillary pore in cement paste [20].

Under insulation curing, see Figure 5b, first, a similar pattern of pore distribution of the cement paste subject to insulation curing is observed compared with that of standard curing. The two-peak mode (derivative curve) and two-stage mode (cumulative curve) is obvious. This similarity indicates the present insulation cover curing performed at very early-age (within 3 days) and does not make and obvious change in the pore structures of the cement paste according to the classification shown in Figure 4. Second, as expected, within the initial three days of insulation curing, the pore volume of the paste is decreased compared with that of the paste under standard curing at the same age. This is consistent with the increase in compressive strength of the concrete after insulation curing. However, at 28 days, the pore volume of the paste subject to insulation curing is increased a little compared with that of standard curing. The total pore volume at 28 days is 0.2530, 0.2812 and 0.2605 mL/g, respectively, of the paste subject to 1, 2 and 3 days insulation curing, respectively. The corresponding pore volume at 28 days of standard curing is 0.2330 mL/g. This indicates that the rising curing temperature (below 35 °C, see Figure 2a) leads to an increase in porosity of the cement paste. Third, according to the classification shown in Figure 4, at 28 days, the capillary pore (V2) and gel pore (V3–V2) volume of the pastes subject to 1, 2 and 3 days insulation in the initial period after casting are (0.142, 0.111); (0.141, 0.141) and (0.149, 0.112) mL/g, respectively. The pore size of d3 is 13.73, 17.1 and 13.72 nm, respectively, which is comparable to that of standard curing.

From the pore distribution result of the paste subject to a constant temperature of 40 °C (LH) in the initial few days after casting (Figure 5c), first, a similar variation pattern to the paste cured under standard (Figure 5a) and insulation curing (Figure 5b) is found. The two-peak mode in the derivative curve and/or two-stage mode in the cumulative curve is displayed. Second, at 28 days, the pore volume of the paste subject to LH curing is increased a little compared to that of standard curing, while it is decreased a little compared to that of insulation curing. The total pore volume at 28 days is 0.2348 mL/g and 0.2476 mL/g, respectively, for the paste subject to 2 days heat curing plus 26 days standard curing (2LH + 26S) and 6 days heat curing plus 22 days standard curing (6LH + 22S). The corresponding pore volume is 0.2330 mL/g, and 0.2530, 0.2812, 0.2605 mL/g for the paste at 28 days under standard curing and insulation curing of 1, 2, 3 days initially that was labeled as (C3 + S25), (C2 + S26), (C1 + S27) in the figures. It should be noted that the total pore volume of the paste after LH curing is lower than that of the paste subject to insulation curing; however, the compressive strength of the concrete subject to LH curing is still lower than the concrete with insulation curing, see Figure 3a. Third, at 28 days, the capillary pore (V2) and gel pore (V3–V2) volume of the pastes subject to 2 and 6 days LH curing in the initial period after casting are (0.136, 0.112) and (0.133, 0.102) mL/g, respectively. The pore size of d3 is 13.72 and 13.73 nm, respectively, which is comparable to that of standard curing too.

As the curing temperature rises to 60 °C (HH), see Figure 5d, first, as expected, the first peak of pore volume variation moves forwards to the direction of the small pore size, and pore volume is decreased compared with the corresponding age of standard curing. A more interesting observation is that with the increase in age, the two derivative peaks are gradually combined together to form a single wider and higher peak, for example at 28 days. This means the stage II and stage III occurring in the cumulative curve is combined together (the slope of the stage III is increased and almost equal to that of stage II). Second, at 28 days, the same as the paste under LH curing, the pore volume of the paste subject to HH curing is increased a little compared to that of standard curing. The total pore volume at 28 days is 0.2464 mL/g and 0.2410 mL/g, respectively, for the paste subject to 2 days heat curing plus 25 days standard curing (3HH + 25S) and 6 days heat curing plus 21 days standard curing (7HH + 21S). However, at 28 days, the capillary pore (V2) and gel pore (V3–V2) volume of the pastes subject to 2 and 6 days HH curing in the initial period after casting are (0.183, 0.058) and (0.192, 0.055) mL/g, respectively. The pore size of d3 becomes about 6.03 nm, which is much smaller than that of standard, insulation and HH curing. Clearly, the volume fraction of the capillary pores is obviously increased for the paste subject to HH curing initially, from 54% to 76% of standard curing to HH curing. The third group of the pore volume (mainly composed of C–S–H gel pores) is greatly decreased (from 46% to 24% of the total pore volume of standard curing to HH curing), although the total pore volume is still comparable to that of standard curing. Principally, formation of denser C-S-H gel with less gel porosity is responsible for the above significant change in the pore composition of cement paste under HH curing (60 °C). 

A similar observation was presented as well in a previous studies; for example, Bahafid [6] found an increase in the capillary porosity and a decrease in the gel porosity by increasing the hydration temperature of well cement paste, especially when the curing temperature is equal to or above 60 °C. This is attributed to a decrease in the C–S–H gel intrinsic porosity and a corresponding increase in C–S–H density for higher curing temperature [6,21,22].

For pastes of C50 concrete (Figure 6) and C80 (Figure 7), a similar variation trend in pore structure, as found as in the paste of C30, is observed for pastes under different curing procedures. The pore characteristic parameters, including total pore volume, capillary pore volume and gel pore volume, as well as the corresponding pore size, are obtained for pastes of C50 and C80 concrete using the method described above. Figure 8a to Figure 8c summarize the pore composition of the pastes of C30, C50 and C80 concrete at 28 days under the four curing procedures (S, C, LH, HH), respectively, in which three curing procedures of insulation curing, C1 + S27, C2 + S26, C3 + S25, and two curing procedures of LH and HH, LH7 + S21, LH3 + S25 and HH7 + S21, HH3 + S25 are included. In addition, normalized average compressive strength with respect to the strength of standard curing is displayed as well to explore the correlation between strength reduction and the curing method used in the initial period after casting. The red line shows the average normalized strength of each curing procedure.

From the pore composition data listed in Figure 8a, first, it can clearly seen that the total pore volume of the paste of C30 concrete is slightly increased after being subject to insulation curing (C), heat curing with relative low temperature (LH) and heat curing with relative high temperature (HH) in the initial period after casting of the concrete compared to that of standard curing (S). Among them, the increase in pore volume of insulation curing group is the highest. However, the reduction in compressive strength at 28 days compared with that of standard curing is the smallest among the three heat curing procedures. This means the total pore volume of the cement paste cannot simply correlate to the macro strength of the concrete in the present case. Second, on the effect of curing procedure on the capillary pore volume of the paste, it can be clearly seen that the capillary pore volume of the paste is monotonously increased from standard curing (20 °C), insulation curing (varied temperature and above 20 °C), low temperature heat curing (40 °C) and high temperature heat curing (60 °C). Especially, for the HH curing, the increase in capillary pore volume is more remarkable than the rest of the heat curing procedures. This variation trend in the capillary pore volume is consistent with the compressive strength of the concrete, although more influencing factors should be involved when concrete strength is considered. The reasons for the strength reduction at 28 days of the concrete subject to heat curing may include not only the effect of pore structure of the paste, but the influence of non-uniform deformation occurring in concrete, especially along the aggregate/matrix interface under varied temperature history within concrete, should be included as well. More detailed studies on the effect of short-term heat curing on long-term mechanical properties of concrete, not only on the microstructure of the cement paste, but also on macro non-uniform deformation of concrete, are continuously needed. 

A quite similar variation trend in the pore composition of the paste of C50 concrete at 28 days under the four curing procedures, see Figure 8b, is observed. First, the total pore volume of the paste of C50 concrete is increased as well after being subject to insulation curing (C), heat curing with relative low temperature (LH) and heat curing with relative high temperature (HH) in the initial period after casting of the concrete compared to that of standard curing (S). As observed in the paste of C30 concrete, the highest increase in pore volume occurred among the curing group of insulation cover curing. However, the reduction in compressive strength at 28 days compared with that of standard curing is the smallest among the three heat curing procedures and is even smaller than that of C30 concrete. Second, the capillary pore volume of the paste is increased as well from standard curing, insulation curing, low temperature heat curing (40 °C) and high temperature heat curing (60 °C). As the paste subject was to higher temperature curing initially (HH), the increase in capillary pore volume is significant relatively. Overall, the variation trend in the capillary pore volume is consistent with the compressive strength of the concrete. For the paste of C80 concrete, see Figure 8c, the total pore volume displays a gradual increasing trend following the order of standard curing (S), insulation curing (C), heat curing with relative low temperature (LH) and heat curing with relative high temperature (HH). This differs from that of C30 and C50 concrete. The capillary pore volume of the paste is increased accordingly, and the increasing amount is relatively obvious as high temperature heat curing (HH) is used. Although the strength reduction due to heat curing is in agreement with the capillary pore content, an exception such as high temperature heat curing is noticeable.

By comparing the results displayed in Figure 8, it can be concluded that the effect of initial curing (increasing curing temperature) used in the initial period after concrete casting on the pore structure of the cement paste mainly presented as the increase in capillary pore fraction. Among the four curing methods used in the present study, the effect of insulation cover curing and low temperature (40 °C) heat curing on the capillary pore content is small compared with that of standard curing, while the effect of high temperature (60 °C) heat curing is significant. This is observed not only in the measurement of the paste of C30 concrete, but also in that of C50 and C80 concrete. This finding may help to understand the effects of heat curing in the initial period after concrete casting on the long-term performance of concrete, and further may help to improve the curing method normally used in pre-casting plants in order to increase very early-age strength of the pre-cast concrete. Obviously, the suggested insulation cover curing is of obvious advantage compared to the traditional heat curing, not only in relation to the cost of pre-cast concrete production, but also in relation to the improvement of long-term performance of concrete.

## 4. Conclusions

In the present paper, an experimental study of the effects of thermal insulation cover on the temperature rise of concrete, and further on the development of mechanical properties of the concrete is performed. One hundred millimeter thick polyurethane foam boards were used for thermal insulation, and three concrete strength grades of C30, C50 and C80, were used as concrete samples. Conventional standard curing, and two heat curing methods with a constant temperature of 40 °C and 60 °C were used for comparison. Pore structures of the cement paste subject to the different curing procedures, including insulation cover curing, heat curing and standard curing, were experimentally measured using mercury intrusion porosimetry. The conclusions obtained in the present study are summarized as following:(1)The temperature of concrete is obviously increased after putting a thermal insulation cover on the surface of the concrete after casting. The temperature rise can not only be observed in high strength concrete (C80), but also in low strength concrete (C30). As 100 mm thick polyurethane foam boards are used and the concrete is cast at room temperature of about 25 °C, t the temperature rise is 10.5 °C, 15.2 °C and 20.3 °C, respectively, for C30, C50 and C80 concrete. The higher the concrete strength, the higher the temperature rise.(2)For C30 concrete, compared with standard curing, the rate of increase in compressive strength due to insulation cover curing is 34%, 22% and 24% at 1, 2 and 3 days, respectively. As expected, initial heat curing of concrete will result in a reduction in long-term strength of the concrete. At 28 days, the strength reduction rate due to the insulation cover curing compared with standard curing is 4.2%, 5.9% and 3.1%, respectively, for insulation curing of 1, 2 and 3 days at the beginning after casting. With respect to the concrete subject to heat curing with a constant temperature of 40 °C and 60 °C initially at the beginning 3 days after casting, the strength reduction rate at 28 days is 8.9% and 10.5%, respectively.(3)For C50 concrete, compared with standard curing, the rate of increase in compressive strength due to insulation cover curing is 24%, 26% and 16%, respectively, after insulation curing of 1, 2, 3 days. At 28 days, the strength reduction rate due to the insulation cover curing compared with standard curing is 0.6%, 3.0% and 2.2%, respectively, for insulation curing of 1, 2 and 3 days at the beginning after casting. With respect to the concrete subject to heat curing with a constant temperature of 40 °C and 60 °C initially at the beginning 3 days after casting, the strength reduction rate at 28 days is 8.6% and 9.1%, respectively.(4)For C80 concrete, compared with standard curing, the rate of increase in compressive strength due to insulation cover curing is 67%, 33% and 23%, respectively, after insulation curing of 1, 2, 3 days. At 28 days, the strength reduction rate due to the insulation cover curing compared with standard curing is −0.8% (increasing), 2.5% and 3.2%, respectively, for insulation curing of 1, 2 and 3 days at the beginning after casting. With respect to the concrete subject to heat curing with a constant temperature of 40 °C and 60 °C initially at the beginning 3 days after casting, the strength reduction rate at 28 days is 4.7% and 5.6%, respectively.(5)Compared with the standard curing, insulation cover curing (within 1–3 days) may also lead to coarsening of the pore structure of the cement paste, which is displayed as an increase in capillary pore fraction. Among the four curing methods used in the present study, the effect of insulation cover curing and low temperature (40 °C) heat curing on the capillary pore content is small compared with that of standard curing, while the effect of high temperature (60 °C) heat curing is significant. This is observed not only inn the measurement of the paste of C30 concrete, but also in that of C50 and C80 concrete.

## Figures and Tables

**Figure 1 materials-15-02781-f001:**
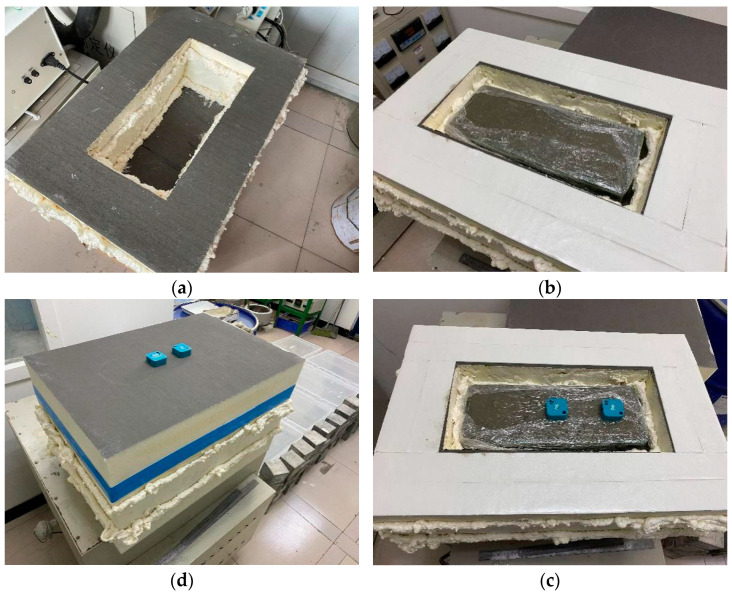
Processes of insulation curing of concrete. (**a**) Prepare insulation box; (**b**) Put specimen; (**c**) Place temperature sensor; (**d**) Glue the top board.

**Figure 2 materials-15-02781-f002:**
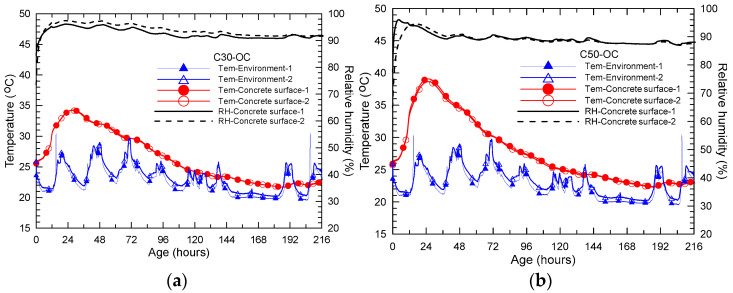
Development of temperature and relative humidity inside the curing box of concrete under insulation conditions (**a**) C30, (**b**) C50 and (**c**) C80.

**Figure 3 materials-15-02781-f003:**
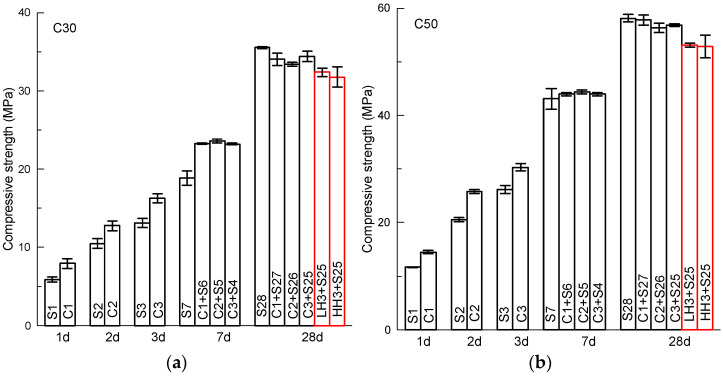
Compressive strength of concrete under different curing procedures (**a**) C30, (**b**) C50 and (**c**) C80.

**Figure 4 materials-15-02781-f004:**
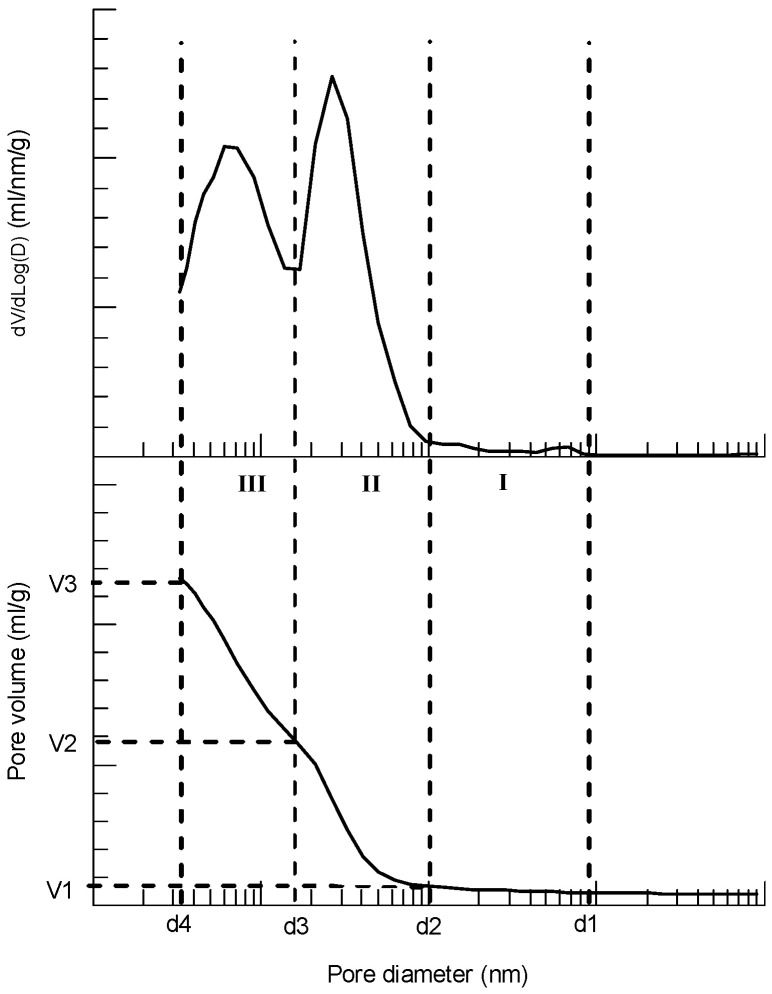
Typical pore distribution of cement paste with three stage divisions.

**Figure 5 materials-15-02781-f005:**
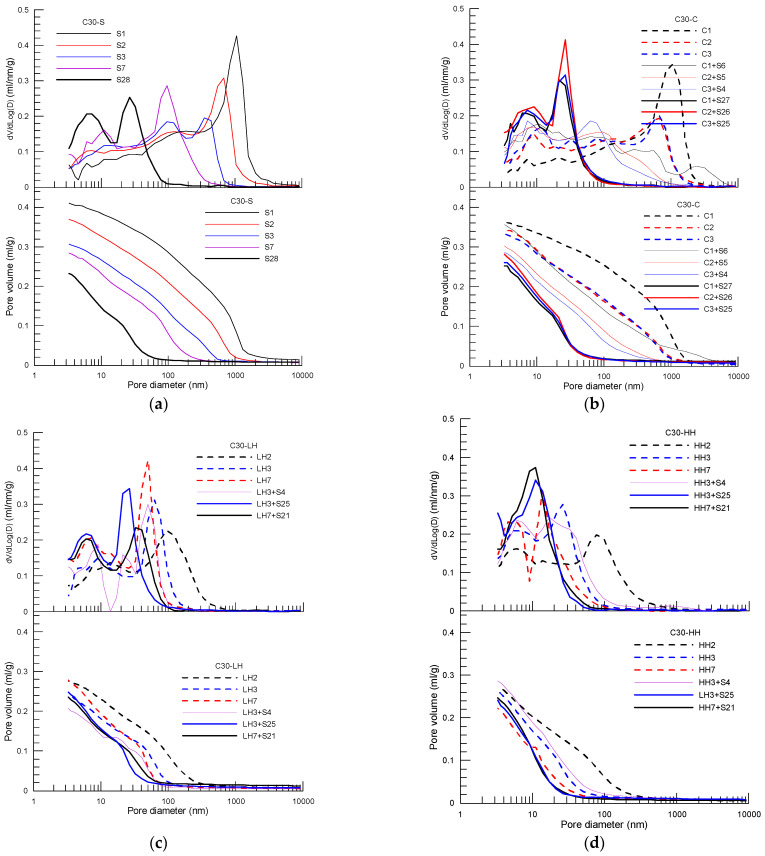
Cumulative (below) and derivative (above) pore volume versus pore diameter curves of the cement paste of C30 concrete subject to different curing procedures: (**a**) standard curing, (**b**) insulation cover curing, (**c**) heat curing with 40 °C and (**d**) heat curing with 60 °C.

**Figure 6 materials-15-02781-f006:**
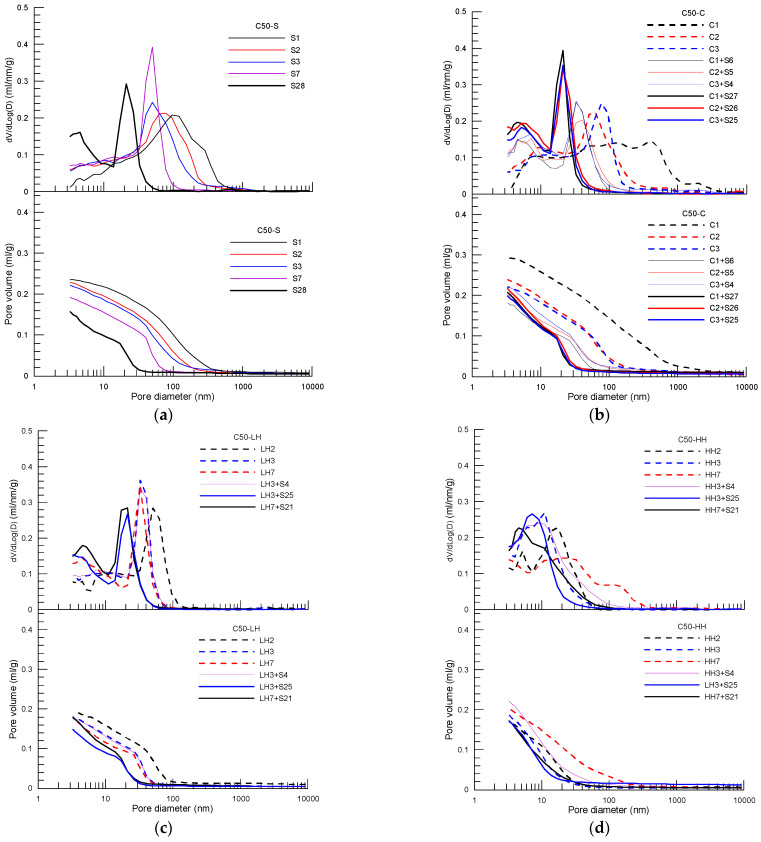
Cumulative (below) and derivative (above) pore volume versus pore diameter curves of the cement paste of C50 concrete subject to different curing procedures: (**a**) standard curing, (**b**) insulation cover curing, (**c**) heat curing with 40 °C and (**d**) heat curing with 60 °C.

**Figure 7 materials-15-02781-f007:**
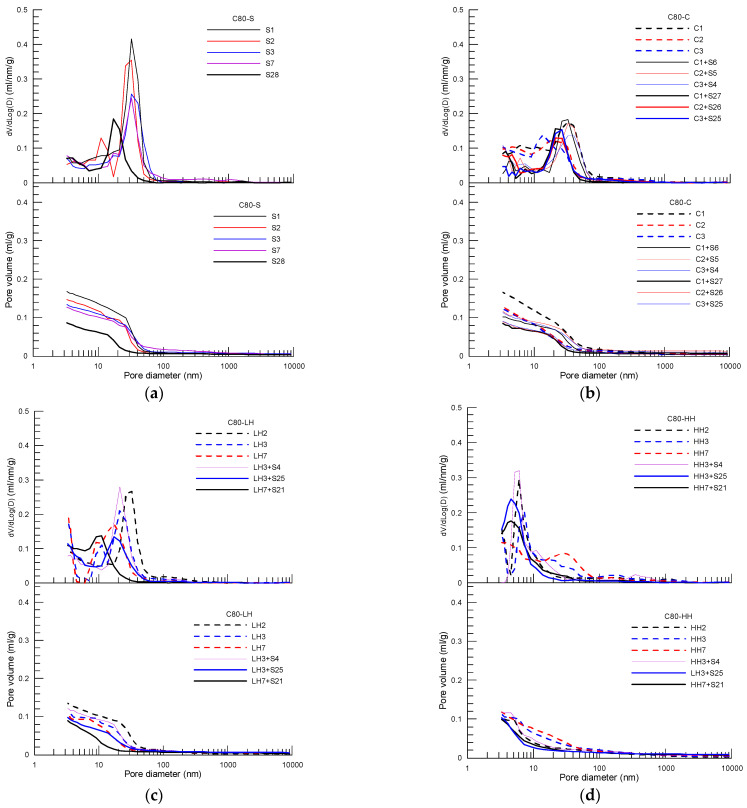
Cumulative (below) and derivative (above) pore volume versus pore diameter curves of the cement paste of C80 concrete subject to different curing procedures: (**a**) standard curing, (**b**) insulation cover curing, (**c**) heat curing with 40 °C and (**d**) heat curing with 60 °C.

**Figure 8 materials-15-02781-f008:**
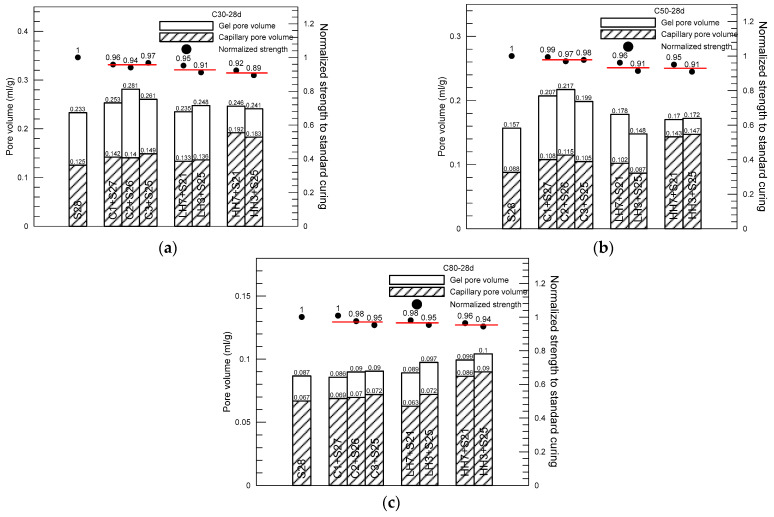
Total, capillary and gel pore volume of the paste of C30 (**a**), C50 (**b**) and C80 (**c**) concrete at 28 days with effect of curing procedure, along with the corresponding normalized strength.

**Table 1 materials-15-02781-t001:** Oxide compositions of cement, silica fume, fly ash and mineral compounds of the cement.

Oxide	Cement			Silica Fume	Fly Ash
	Percentage	Compound	Percentage		
CaO	60.28	C_3_S	49.58	0.81	5.08
SiO_2_	23.47	C_2_S	28.04	90.56	47.02
Al_2_O_3_	7.41	C_3_A	7.28	0.41	35.06
Fe_2_O_3_	2.97	C_4_AF	8.57	0.52	3.88
MgO	1.97	−	−	0.95	1.36
K_2_O	0.62	−	−	1.59	1.30
Na_2_O	0.14	−	−	0.63	1.18
SO_3_	2.39	−	−	−	0.89
Loss	2.8	−	−	3.72	1.85

**Table 2 materials-15-02781-t002:** Mix proportions of concrete, kg/m^3^.

No.	Cement	Silica Fume	Fly Ash	Water	Sand	Stone	W/B
C30	240	-	60	186	750	1150	0.62
C50	345	-	85	185	685	1090	0.43
C80	467	52	-	156	602	1184	0.30

## Data Availability

The data presented in this study are available on request from the corresponding author.

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
