# Peer review of "Effect of the Thermal Insulation Cover Curing on Temperature Rise and Early-Age Strength of Concrete"

_materials, 2022, doi:10.3390/ma15082781_

Round 1

Reviewer 1 Report

The article talks about the effect of curing regimes on the strength gain and porosity/pore structure of three different concrete mixtures. There are issues with the language, and it is hard to follow the manuscript. The discussion sections are long and confusingly worded. It is recommended that the authors get this manuscript proofread and corrected by a native English speaker. The major comments are listed below. The manuscript can be considered only after addressing these major revisions –

P1, L15 - Please rephrase this sentence. This is confusing.

P1, L13 and 17 - Suffering - subjected

P1, L18-22 - Confusing. Please rephrase.

P3, L96 - The authors are trying to compare a mixture containing fly ash to a mixture containing SF. Using SF provides high strength at early ages compared to fly ash irrespective of the curing condition, it does not allow for a one-to-one comparison with fly ash even at elevated temperature curing. Further, the binder content in all mixtures is different. If the authors wanted to vary the water to binder ratio, the water content should have been varied for C30 and C50 mixtures. Comparing these mixes with each other does not yield results that would make complete sense. The authors may please justify their reasoning for the choice of mixtures.

Multiple locations throughout the paper - use the degree symbol before the temperature.

P5, Fig. 2 - Was there a study of the internal concrete temperature for these samples? The surface temperature will equilibrate with the environment but depending on the member thickness, the internal temperature will continue to cure the sample. How is this accounted for?

P7, L 222-235 - This is expected since you use silica fume. Silica fume mixes show higher strength in general at all ages due to enhanced hydration and pozzolanic reaction. This is also increased further due to temperature increase. The authors may please comment on what would happen if fly ash were to be used for the low w/c concrete also?

Fig. 5,6,7 - Please consider presenting the most important information (i.e. threshold pore diameter.) in the form of a bar chart or a scatter plot for easy comparison for all concrete classes tested. All these curves are not necessary. This whole curve can be fit into three plots and made a lot less confusing. This is hard to follow in the current state. 

P8-15- The explanation for MIP is long and hard to follow in places. It is recommended that the authors shorten the discussion on MIP section. There appears to be quite some overanalyzing on the part of the authors to convey 2-3 points which can be grouped into -

  1. Change in threshold pore diameter with age and curing regime,
  2. Change in capillary porosity and gel porosity with curing regime,
  3. Correlation of porosity to strength results.

Author Response

Responds to reviewers’ comments

Dear Editor of Materials;

The paper, materials-1674003, entitled “Effect of the Thermal Insulation Cover Curing on Temperature Rise and Early-Age Strength of Concrete was carefully revised according to reviewers’ comments and suggestions from the editor. Firstly, authors would like to express deep appreciation to reviewers and editor for providing useful suggestions and comments. Below is our responds to the comments.

Comments and Suggestions for Authors
-Reviewer 1

The article talks about the effect of curing regimes on the strength gain and porosity/pore structure of three different concrete mixtures. There are issues with the language, and it is hard to follow the manuscript. The discussion sections are long and confusingly worded. It is recommended that the authors get this manuscript proofread and corrected by a native English speaker. The major comments are listed below. The manuscript can be considered only after addressing these major revisions.

Yes. The paper was revised carefully.

P1, L15 - Please rephrase this sentence. This is confusing.

Yes. Please note in order to remold fast in pre-cast concrete plan, the concrete members subject to a heat curing after casting 1-2 days normally. The thermal insulation curing is like to replace the heat curing in pre-cast concrete production. Therefore, the developed thermal insulation cover curing is taken place after concrete casting immediately for 1-3 days.

This sentence is revised as “The test results show that the increasing rate of compressive strength comparing with standard curing is 22%-34%, 16%-26, 23%-67% respectively for C30, C50, C80 concrete after subjecting 1-3 days thermal insulation cover curing at initially period after concrete casting.

P1, L13 and 17 - Suffering - subjected

Yes. It was revised.

P1, L18-22 - Confusing. Please rephrase.

Yes. Please note after concrete subjecting heat curing at initial period after casting in order to remold fast, the strength at longer age, such as at 28 days, should reduce comparing with that of standard curing within all 28 days. This fact is well known for concrete production in pre-cast plant. If know this fact, you may understand the conclusions described in lines 18-22.

P3, L96 - The authors are trying to compare a mixture containing fly ash to a mixture containing SF. Using SF provides high strength at early ages compared to fly ash irrespective of the curing condition, it does not allow for a one-to-one comparison with fly ash even at elevated temperature curing. Further, the binder content in all mixtures is different. If the authors wanted to vary the water to binder ratio, the water content should have been varied for C30 and C50 mixtures. Comparing these mixes with each other does not yield results that would make complete sense. The authors may please justify their reasoning for the choice of mixtures.

Thanks for the comments. We are not comparing the effect of fly ash or silica fume addition on compressive strength, while, we are use the three practically used concrete mixtures in pre-cast concrete plant, to investigate the effect of thermal insulation curing on their early-age strength. The reason why selecting three concrete mixtures, is simply to find the overall efficiency of the thermal insulation curing on the different concrete types used in practice. It is not the focuses to study the effect of fly ash or silica fume in the mixture. Therefore, we are mainly comparing the results between the different curing procedures.

Multiple locations throughout the paper - use the degree symbol before the temperature.

Yes. The error was corrected in the revised version.

P5, Fig. 2 - Was there a study of the internal concrete temperature for these samples? The surface temperature will equilibrate with the environment but depending on the member thickness, the internal temperature will continue to cure the sample. How is this accounted for?

No. As starting, we just measured the temperature of concrete surface. Yes, you are right, the inside temperature of concrete may differ due to heat transformation, and, it may further influence by the thickness of the members. More detailed study on the temperature filed is doing now, including the possible temperature gradient before and after thermal insulation curing. 

P7, L 222-235 - This is expected since you use silica fume. Silica fume mixes show higher strength in general at all ages due to enhanced hydration and pozzolanic reaction. This is also increased further due to temperature increase. The authors may please comment on what would happen if fly ash were to be used for the low w/c concrete also?

Thanks for the comments. Please note for each type of concrete, the principal target of the present study is to investigate the effect of thermal insulation curing on very early age strength. The temperature effects are the main consideration of the study. The addition of silica fume and fly ash in the corresponding concrete (C80 and C50, C30 respectively) is just the needs on the concrete type. For example, silica fume can increase the strength of the concrete, which was used in C80 concrete. On the other hand, the use of silica fume in concrete with low water to binder ratio can greatly improve the viscosity of the fresh mixture. This is another more important reason in practice than increasing strength. Fly ash normally used in low strength concrete (C30 and C50 in the present study) to improve the workability of the fresh concrete. The addition of fly ash addition will reduce early-age strength of concrete. Therefore it is few be used in high strength concrete along, however, if it is combined with other finer additives, such as silica fume, it can be used in relative high strength concrete as well. All depend on the needs.

Fig. 5,6,7 - Please consider presenting the most important information (i.e. threshold pore diameter.) in the form of a bar chart or a scatter plot for easy comparison for all concrete classes tested. All these curves are not necessary. This whole curve can be fit into three plots and made a lot less confusing. This is hard to follow in the current state.

Thanks for the comments. The purpose of Figs. 5 to 7 is just to show the possible change on the pore structures of the paste subjecting to different curing procedures. They are clearly showed that high temperature curing, i.e. higher than 60 0C, will increase the content of capillary pore fraction. This is sufficient to explain what happens as curing with a higher temperature on the paste. Certainly, as described, it may still not fully explain the mechanisms of the strength reduction at 28 days after subjecting a heat curing on concrete at initial period. More detailed study on this issue is needed.

According to our test results, the so-called threshed pore diameter, such as pore size corresponding to the peak pore volume changing, cannot correlate with the strength variation. Therefore, it is not necessary to show the so-called threshold pore diameter in the figures.  

P8-15- The explanation for MIP is long and hard to follow in places. It is recommended that the authors shorten the discussion on MIP section. There appears to be quite some overanalyzing on the part of the authors to convey 2-3 points which can be grouped into -

It is very important to explain and use the pore test results, for example how to distinguish capillary pore and C-S-H gel pore from the test data. This is why we use a long paragraph to explain the pore data obtained (Fig.4). According to described classification, the test results are analyzed with relative detailed, which are necessary.

Change in threshold pore diameter with age and curing regime,

Please see explain above.

Change in capillary porosity and gel porosity with curing regime,

Fig.8 displays the relationship of capillary and gel pore volume fraction and curing regime.

Correlation of porosity to strength results.

It was showed in Fig.8, in which the relation between total porosity and the strength reduction normalized with standard curing is showed. It looks without direct relation between strength and porosity of the paste, while more interesting relationship of capillary pore volume and strength reduction is observed.

Submission Date

25 March 2022

Date of this review

01 Apr 2022 06:45:42

Reviewer 2 Report

Attractive manuscript related to “the effects of thermal insulation cover curing on the temperature rise of concrete”.

In general, the paper is well presented, but some information shall be clarified or corrected/completed:

  1. Line 45 and other: please replace “oC” with “ºC”;
  2. Section “2.1 Materials and mix proportions”: you can include some information on Cement, Silica Fume, Fly Ash, Sand and Stone (origins and properties);
  3. Line 93: what kind of “superplasticizing” was used?
  4. Line 124: please correct “… temperature of 20 20C; …”;
  5. Figure 2: in the graph (c), why is the difference between “Tem-Environment-1 and -2” are smaller than the differences observed in graphs (a) and (b)?
  6. Figure 3 and other: the numbers written over the bars are not very readable. Try cutting the weave at the bottom of the text;
  7. Figure 4: the values (or variables) on the ordinate axis (dV/dLogD) are missing. Is that correct?
  8. Figure 8: please explain the red line meaning (average?);
  9. Section “4. Conclusions”: the use of graphic trends may make some conclusions more quickly understandable;
  10. It seems appropriate to include a list of acronyms (or “Abbreviations”) before “References”;
  11. In “References”: please include the digital object identifier (DOI) for all references where available;
  12. There are many old references. Try to analyse and include more recent ones;
  13. It may be appropriate to give some indication of the cost difference that this procedure (with polystyrene foam) can have on full-scale work compared to other solutions used for curing concrete.

Author Response

Comments and Suggestions for Authors
-Reviewer 2

Attractive manuscript related to “the effects of thermal insulation cover curing on the temperature rise of concrete”.

In general, the paper is well presented, but some information shall be clarified or corrected/completed:

Line 45 and other: please replace “oC” with “ºC”;

Yes. These errors are corrected.

Section “2.1 Materials and mix proportions”: you can include some information on Cement, Silica Fume, Fly Ash, Sand and Stone (origins and properties);

Yes. The related materials, such as cement, fly ash and silica fume, and their basic properties are added. Please see lines 92-95 and Table 1 in the revised version.

Line 93: what kind of “superplasticizing” was used?

The information of the superplasticizer was added. See line 100 of the revised version.

Line 124: please correct “… temperature of 20 20C; …”;

This error is corrected. It should be “20±2 0C”.

Figure 2: in the graph (c), why is the difference between “Tem-Environment-1 and -2” are smaller than the differences observed in graphs (a) and (b)?

As described in the text, two temperature sensors are used to measured environmental temperature (testing room). Therefore, there are two curves of the environmental temperature for each testing period. They are looks reasonable.

Figure 3 and other: the numbers written over the bars are not very readable. Try cutting the weave at the bottom of the text;

Yes. They are revised.

Figure 4: the values (or variables) on the ordinate axis (dV/dLogD) are missing. Is that correct?

Fig.4 is an illustration figure to show the classification of pores in the cement paste. Therefore, it is not necessary to show the value.

Figure 8: please explain the red line meaning (average?);

Yes. It is average value of the normalized compressive strength.See lines 403-404 of the revised version.

Section “4. Conclusions”: the use of graphic trends may make some conclusions more quickly understandable;

The conclusion of point 2 is too long to read. It is revised into 3 short points now.

It seems appropriate to include a list of acronyms (or “Abbreviations”) before “References”;

In “References”: please include the digital object identifier (DOI) for all references where available;

The references are listed according to the requirements of the journal.

There are many old references. Try to analyse and include more recent ones;

The references are cited according to the relevance of comments or points described in the paper, not to the published date. The references used in the present paper are reasonable.

It may be appropriate to give some indication of the cost difference that this procedure (with polystyrene foam) can have on full-scale work compared to other solutions used for curing concrete.

Thanks. It is a good point. The analyses must be provided as long as the method is fully mastered. However, present work is only an experimental study on the developed curing method. The economic analyses should be provided after obtain more data, especially after a typical application of the method.

Submission Date

25 March 2022

Date of this review

31 Mar 2022 22:13:47

Round 2

Reviewer 1 Report

The authors have addressed most comments raised. Considering the fact that these mixtures are "practical" mixtures, the changes made are acceptable.

The paper can be accepted in its current format.